# The GABA Polarity Shift and Bumetanide Treatment: Making Sense Requires Unbiased and Undogmatic Analysis

**DOI:** 10.3390/cells11030396

**Published:** 2022-01-24

**Authors:** Yehezkel Ben-Ari, Enrico Cherubini

**Affiliations:** 1Neurochlore, Batiment Beret Delaage, Campus Scientifique de Luminy, 13009 Marseille, France; 2European Brain Research Institute (EBRI)-Rita Levi-Montalcini, 00161 Roma, Italy; cher@sissa.it

**Keywords:** autism spectrum disorders, intracellular chloride levels, bumetanide, development, brain disorders, GABA polarity actions

## Abstract

GABA depolarizes and often excites immature neurons in all animal species and brain structures investigated due to a developmentally regulated reduction in intracellular chloride concentration ([Cl^−^]_i_) levels. The control of [Cl^−^]_i_ levels is mediated by the chloride cotransporters NKCC1 and KCC2, the former usually importing chloride and the latter exporting it. The GABA polarity shift has been extensively validated in several experimental conditions using often the NKCC1 chloride importer antagonist bumetanide. In spite of an intrinsic heterogeneity, this shift is abolished in many experimental conditions associated with developmental disorders including autism, Rett syndrome, fragile X syndrome, or maternal immune activation. Using bumetanide, an EMA- and FDA-approved agent, many clinical trials have shown promising results with the expected side effects. Kaila et al. have repeatedly challenged these experimental and clinical observations. Here, we reply to the recent reviews by Kaila et al. stressing that the GABA polarity shift is solidly accepted by the scientific community as a major discovery to understand brain development and that bumetanide has shown promising effects in clinical trials.

## 1. GABA Polarity Shift: A Brief Historical Perspective

In 1989, recording from hundreds of neonatal rat hippocampal slices, we made three completely unexpected observations [1].

Instead of the expected membrane hyperpolarization, in neonatal neurons, GABA mediates synaptic currents, and exogenous GABA applications induce a membrane depolarization that often reaches the threshold for action potential generation. The depolarizing/excitatory action of GABA progressively shifts with age toward a hyperpolarizing/inhibitory action.

This early developmental alteration is due to a higher intracellular chloride concentration ([Cl^−^]_i_) that is progressively reduced following the same time course as the excitation/inhibition shift.

The interplay between the depolarizing action of GABA and glutamate generates a primordial form of polysynaptic synchrony, crucial for synaptic wiring and refinement of local neuronal circuits that we termed “giant depolarizing potentials”, or GDPs. GDPs disappear when the polarity of GABA shifts from the depolarizing to the hyperpolarizing direction.

Collectively, these observations (Figure 1) indicate that the actions of GABA follow a developmental course with a significant polarity shift of its actions, and this underlies major changes in cell and network activity. Earlier studies had observed a late maturation of inhibition [2,3,4], but this study was the first to put together a global picture of the developmental actions of GABA and to highlight the central role of [Cl^−^]_i_ in these events. Since then, the GABA developmental polarity shift became a major topic of investigation, providing considerable advances in understanding its underlying mechanisms and biological significance, notably in relation to the trophic actions of GABA [5,6,7,8,9]. This phenomenon is well preserved throughout evolution, as suggested by its presence in virtually all brain regions and animal species investigated. However, the mechanisms underlying the developmental excitatory/inhibitory shift of GABA were still unknown. Ten years later, Rivera et al. confirmed this shift and made a significant contribution to the field showing that it results from the differential temporal expression of the cation–chloride cotransporters NKCC1 and KCC2, involved in chloride uptake and extrusion, respectively [10]. This is a fitting example of how science proceeds to reach an objective.

However, Kaila et al. have recently written reviews in *Cells*, *TINS*, *Epilepsia*, and bioxRiv presenting a very biased summary of these facts, omitting quite systematically to refer to our discovery and quoting much later works published by their groups. Using the GDP terminology while referring to their own papers goes beyond a misquotation issue [12]. Our first motivation here was simply to stress these unquestionable facts. More importantly, our aim is to challenge what we consider a dogmatic biased view on what has been or ought to be carried out to understand the GABA polarity shift and its experimental or clinical implications.

## 2. Fact 1: The Polarity Shift Is Not Anymore Debated

Since our original paper, the developmental shift of GABA from the depolarizing to the hyperpolarizing direction has been observed at early developmental stages in several brain regions in rodents using in vitro recordings. It has been detected also in newborn neurons in the adult brain. The few studies that have challenged this shift are based on wrong assumptions and irreproducible results, as stressed by a review signed by most of the experts of the domain [13] and by a parallel study by Kaila et al. [14]. In fact, the most reliable and direct evidence of the GABA shift stems from non-invasive single GABA channel recordings to determine E_GABA_ and NMDA-receptor-mediated currents to determine V_rest_. All other techniques are invasive and provide indirect measures of these parameters.

In vivo studies, including those made by Kaila et al., are derived from recordings in anesthetized animals [15,16,17]. Conclusions from these studies are hampered by the effects of anesthetics (reviewed in [13]. Thus, in vivo and in vitro, anesthetics impact heavily dendritic excitability and propagation of calcium currents [18]. A recent study performed on non-anesthetized rodents has confirmed the excitatory actions of GABA at least on pyramidal neurons of the hippocampus in vivo [19] (Figure 2). 

Therefore, stating as in [17] that “the very existence of the developmental shift in [Cl^−^]_i_ described above has been intensely debated because of the lack of such (adequate) measurements” referring to our collective paper where these “challenges“ are shown to be artifacts [13] is misleading and infirmed by the published evidence. Clearly, recognizing the limitations of each technique is a very useful exercise and combining in vitro and in vivo preparations and techniques are indispensable to reduce, at least in part, these limitations. Briefly, the polarity shift is a fact. What remains is to better understand all its facets. Kaila et al. have also challenged our earlier observations on the oxytocin-mediated neuroprotective shift during birth, suggesting that GABA remains depolarizing during birth [20,21]. There are important differences between the two experimental conditions [22]. However, most importantly, the tools used by Kaila et al. to determine [Cl^−^]_i_ levels are indirect and unreliable. At birth, many rodent CA1 pyramidal neurons have little or no dendrites, and therefore, applying caged GABA on dendritic regions is difficult to interpret [23]. Additionally, determining chloride extrusion capacity by photolysis of whole-cell recordings neither provides a reliable estimation of DF_GABA_ nor the dynamics of [Cl^−^]_i_ levels. All invasive recordings of immature neurons alter V_rest_ by over 15 mV [24], impacting the determination of E_GABA_. Single-channel, cell-attached recordings remain the only reliable technique, in particular in immature neurons. It also bears stressing that the GABA shift during birth was also abolished in rodent models of Rett syndrome [25], autism spectrum disorders, fragile X syndrome [26], and maternal immune activation [27]. It bears stressing that oxytocin controls the GABA excitatory-to-inhibitory shift (via KCC2 activation) [28], plasma oxytocin levels are lower in children with ASDs [29], sensitive parenting is impacted by oxytocin [30], and intranasal oxytocin attenuates social deficits in patients with ASDs [31].

## 3. Fact 2: Reducing High [Cl^−^]_i_ Levels with Bumetanide Has Beneficial Effects in Several Neurological and Psychiatric Disorders

In a wide range of experimental conditions and disorders, the shift of GABA from the hyperpolarizing to depolarizing direction is absent or delayed with high [Cl^−^]_i_ levels and excitatory actions of GABA, notably in epilepsies [32,33,34] (reviewed in [35]). Similar high [Cl^−^]_i_ levels have been observed in many brain disorders [35]. Bumetanide, an NKCC1 chloride importer antagonist, has raised considerable interest since it is able to restore the inhibitory action of GABA in vitro and attenuate the severity of the disorders in animal models extending from certain types of epilepsies, autism spectrum disorders (ASDs), Parkinson’s disease, brain trauma, Down syndrome, chronic pain, and even glioblastoma, to quote only a few of a long list [35]. Attempts to cause the same effect using a KCC2 activator have, to the best of our knowledge, failed most likely because of the short life span of the protein and its rapid internalization after insults [36]. A recent study has shown that bumetanide is the most promising among over 1300 widely used treatments blindly tested to reverse APOE specific genetic signatures. Furthermore, patients over 65 years old who were treated with bumetanide had a much lower likelihood to have Alzheimer’s disease, and bumetanide reduced in mice physiological and morphological features of Alzheimer’s disease [37].

Relying on these observations, clinical studies using oral administration of bumetanide has been found to mitigate the severity of ASD symptoms in hundreds of children and adolescents in many independent trials made in France, Sweden, Tunisia, The Netherlands, and China [21,37,38,39,40,41,42,43,44,45,46] (Figure 3 and Table 1). 2 meta-analysis of studies suing bumetanide to treat autism reveal that bumetanide attenuates the severity of ASDs (496 children) [47,48]. To the best of our knowledge, only one trial by Bruining et al. has shown negative results with no significant differences between treated and placebo [45]. However, the authors stress in that study that bumetanide did significantly attenuate the repetitive behavioral scale, reflecting the usefulness of the treatment. In a later study, the same group showed that bumetanide attenuates ASDs in patients having special EEG features, in fact these allow the prediction of treatment outcomes, illustrating the importance of identifying responders using specific biomarkers [49] (also see below).

Unfortunately, the large phase III trial performed by Servier and Neurochlore failed to reach significant differences between bumetanide treated and placebo (211 children 2–7 years old and 211 adolescents 7–18 years old, in 40 centers in Europe, USA, Brazil, and Australia). Yet, several hundreds of children with ASDs have been and are treated successfully with bumetanide (Table 1) [48]. This failure illustrates the heterogeneity of ASDs, suggesting that a single treatment to all patients with ASDs might be difficult, which makes the identification of subpopulations of bumetanide responders indispensable. This, however, does not invalidate the efficacy of bumetanide but does raise important questions as to the biomarkers of the responders to the treatment. To this end, we are currently reexamining all the data using artificial intelligence to identify subpopulations of responders.

Compatible with an excitatory action of GABA [52], bumetanide was also able to reduce the paradoxical effects of benzodiazepine on both cognition and EEG in a girl affected by ASD, epilepsy, and cortical dysplasia [53]. Visual eye tracking and brain imaging techniques have confirmed the beneficial actions of oral bumetanide in ASD adolescents [51,52]. In addition, bumetanide attenuates the autism traits but not the seizures in patients with tuberous sclerosis [50]. Bumetanide attenuated social behavior, reduced irritable and hyperactive behavior, and health-related quality of life. Additionally, atypical event-related potentials (ERPs) were attenuated by bumetanide, providing an electrical partial signature of the amelioration. More recently, Bruining et al. have shown that the efficacy of bumetanide in patients with ASDs can be predicted relying on EEG measures [49] (Figure 4). These patients had unique EEG features, and bumetanide administration restored the EEG and attenuated the clinical symptoms of ASDs. Interestingly, the effects of bumetanide could be predicted by the EEG configuration. Collectively, these studies suggest that biological and/or genetic markers are essential to identify a subpopulation of patients for whom bumetanide treatment will be effective. It seems a priori difficult to consider these ameliorations as “illusions”, as suggested by Kaila and Löscher [54] or being exclusively peripherally mediated.

## 4. Fact 3: The Side Effects of Bumetanide Are Well Controlled and Limited

Yet, relying on the postulate that bumetanide has strong ototoxic side effects, Löscher and Kaila have repeatedly advocated against the usefulness of bumetanide to treat brain disorders [54,55,56]. Recently, Kaila et al. pleaded against pursuing phase III clinical trials in spite of promising preliminary observations stating: “Soul et al. [37] concluded that definitive proof of bumetanide’s efficacy awaits an appropriately powered Phase 3 trial, *which we would emphatically advise against because of the many reasons explained in this commentary*.” The authors stress two major limitations of bumetanide: ototoxicity and poor blood–brain barrier (BBB) permeability. Concerning ototoxicity, Kaila et al. refer systematically to the trial of Pressler et al. (2015), which consisted in treating 2-day-old babies affected by encephalopathy with bumetanide (four i.v. injections), antiepileptics (phenobarbital, midazolam, phenytoin, and lidocaine) and antibiotics (tobramycin or gentamycin), known to have ototoxic effects [57]. Pressler et al. stress consistently that seizures associated with neonatal encephalopathies (three babies died during the trial) cannot be compared with other types of more adult forms of epilepsies [58]. A recent trial made by the Boston group [37] has also indicated the limited side effects of bumetanide, in addition to phenobarbital, to treat seizures in babies. There is little doubt that speculating from life-threatening encephalopathy with all the major metabolic disturbances to ASDs or other disorders where bumetanide is administered orally (0.5 or 1 mg) is, at best, a lack of understanding of the issues in the development of treatments and ethically unacceptable. Incidentally, to the best of our knowledge, there is not a single reference showing ototoxic actions of bumetanide following *oral treatment* with bumetanide. Therefore, the assumption that bumetanide should not be used to treat neuropsychiatric disorders is scientifically biased and unsubstantiated. It is not astonishing that the administration of bumetanide is acceptable by authorities all over the world precisely because of the lack of severe uncontrollable side effects (usually diuresis and hypokalemia). Admittedly, it is impossible to demonstrate that [Cl^−^]_i_ levels are high in central neurons of children with ASDs, yet the convergence of experimental and human data is not that frequently reached in the development of treatment of brain disorders. 

The issue of the BBB permeability deserves indeed a detailed reply beyond the scope of this rebuttal. The vision of a static BBB identical in immature, adults, rodents, humans in health and disease is somehow outdated. It is important to take into account the high complexity of the BBB and its dependence on species, age, local permeability changes in disorders, etc. [59,60]. Reduction of BBB effectiveness are well characterized in many central and peripheral disorders [61]. We must admit that data on human BBB in chidren with ASD notably is completely lacking. Is it completely excluded that the BBB is altered in children with ASDs, and an active transport occurs? Is it reasonable to consider that measures in rodents represent babies, adults, and patients with Parkinson, ASD, Alzheimer etc? there are now extensive investigations showing the profound alterations of the BBB in many disorders in humans [60]. The recent studies showing that bumetanide attenuates both EEG alterations in patients with ASDs and clinical manifestations cannot be readily reconciled with an exclusive peripheral action [49]. A peripheral type of action of bumetanide likely contributes. Thus, GABA and muscimol stimulate the release of adrenocorticotropic hormone (ACTH) or the growth hormone (GH) in pituitary cells. Consistent with a depolarizing action of GABA, RT–PCR analysis from cultured anterior pituitary cells (obtained from adult female rats) unveiled high levels of NKCC1 but not KCC2 mRNA [62]. Bumetanide may act on these cells by reducing serum levels of ACTH, GH, and cortisol, which, in autistic subjects, were found to be higher, compared with controls [63]. The recent study showing that out of 1300 widely used agents tested, bumetanide is the most promising drug to treat Alzheimer [12] illustrates the usefulness of repositioning this agent. A large study on millions of seniors over 65 years old revealed a significance decreased incidence of Alzheimer in those who used Bumetanide [12]. To summarize, the experimental and clinical data accumulated cannot readily be reconciled by exclusively peripheral action, and therefore, a global, central, and peripheral action is more likely. Additionally, independently of the exact site of action, bumetanide ameliorates the life of infants with seizures or ASDs, and, from an ethical point of view, it remains an important goal to pursue.

Our phase III trial was accepted by the European Medicine Agency precisely because the side effects are limited—“Primum non Nocere”—and controllable, the health authorities requiring a positive benefice/risk ratio. If KCC2 activators will be found in clinical trials to attenuate brain disorders, this would be a highly recommended avenue, even if they have a short life span. In addition, the dogma that KCC2 activators are more suitable, because contrary to NKCC1, KCC2 is exclusively neuronal is challenged with the demonstrated presence of KCC2 on pancreatic cells that modulate insulin release [62]. Dogmas are always open to challenge! It is perhaps safe to state that there are, to the best of our knowledge, no successful advanced phase II or III trials on any brain disorder centered on KCC2 -targeting molecules, in spite of the immense investments by the pharmaceutical industry. Therefore, even if bumetanide attenuates a proportion of children with ASDs, this remains an important contribution to the treatment of this disorder that has remained refractory to treatments. The actual trend of the pharmaceutical industry is to develop novel bumetanide analogs, and this attests to the importance of the pioneer works and trials we and others have made.

## 5. Conclusions: Future Directions Will Have to Rely on Many Avenues

When facing such a complex domain with wide experimental and clinical implications, there are many important avenues that deserve investigation. Kaila et al. insist on their reviews on the fine regulation of NKCC1 and KCC2, including single-cell approaches, to determine the fine local regulation of cotransporters. Nevertheless, they also stress that determining these mechanisms and the amount of total mRNA of the cotransporter do not directly transfer to functional efficacy [36]. At the end of the day, determining [Cl^−^]_i_ levels remains an unavoidable step. Additionally, there are many other avenues that are indispensable, depending on the question being asked. We advocate, for instance, more investigations on what occurs in utero and during birth in disorders such as ASDs, maternal immune activation, and genetic disorders such as Rett syndrome. As ASDs are “born” in utero and birth, we should be able to identify already at birth babies susceptible to be diagnosed later with ASDs. Indeed, this is feasible using machine learning [64]. The next step would be to identify the sequels of the inaugurating insult and the deviations it produces in brain construction. The “neuroarcheology “ concept posits that these early insults lead to the persistence of neurons endowed with immature features that are the cause of the disorder [65]. This concept will remain fundamental to understand, diagnose early, and treat ASDs and many other disorders. Clearly, the heterogeneity and dynamic features of developmental processes necessitate a better understanding of the deviations produced by the initial insult and whether and how this impacts parturition and birth. This has major implications and has remained terra incognita for a long period. In a few words, there are many ways to reach the grail, and many grails to uncover.

## Figures and Tables

**Figure 1 cells-11-00396-f001:**
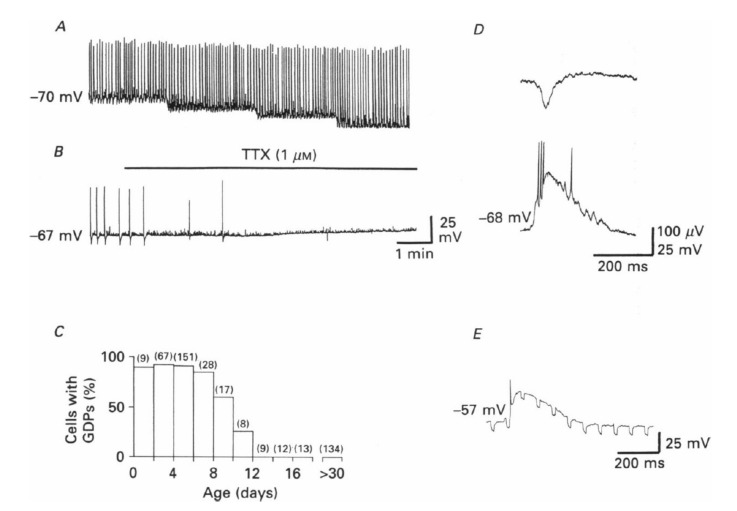
Giant depolarizing potentials (GDPs) and depolarizing actions of GABA in immature hippocampal pyramidal neurons. This is the first description of GDPs, which are synaptic events readily blocked by TTX (**A**,**B**). They are present during the first 12 days of post-natal life in rodents (**C**); (**D**) concomitant extracellular (**upper trace**) and intracellular (**lower trace**) recordings of a GDP; (**E**) spontaneous GDPs associated with an increase in input conductance, as shown by changes in electrotonic potentials resulting from injection of constant hyperpolarizing current pulses (−200 pA) through the recording electrode. From [11].

**Figure 2 cells-11-00396-f002:**
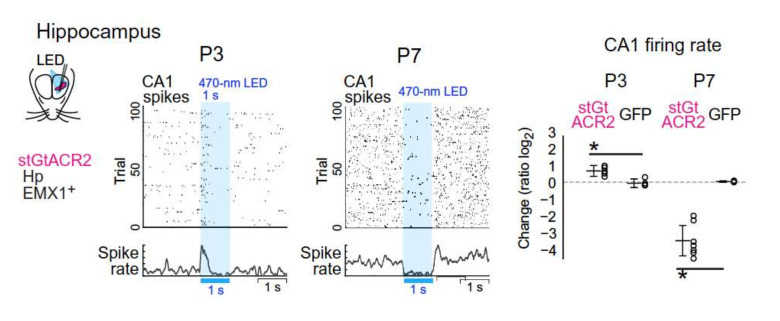
GABA excites pyramidal neurons in vivo in unanesthetized rodents. Changing the role of anion conductance by photostimulation of channelrhodopsin virally expressed in the soma of non-GABAergic neurons generates different effects at P3 and P7. This stimulation increased the firing rate at P3 and reduced it at P7. With permission From [19].

**Figure 3 cells-11-00396-f003:**
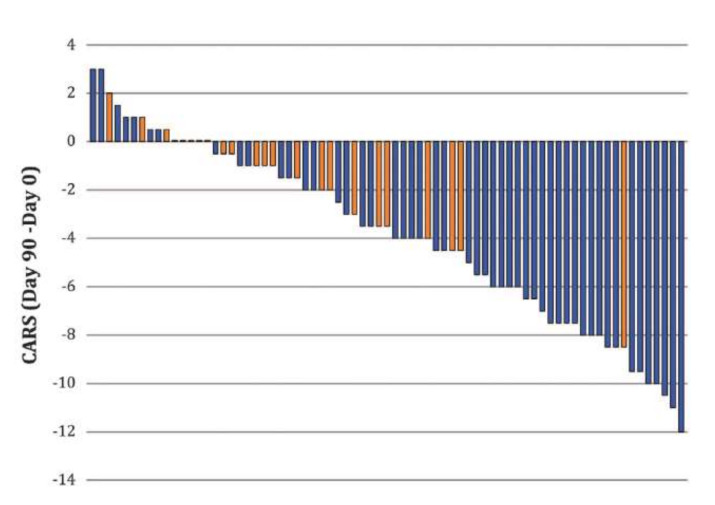
Bumetanide attenuates the severity of autism spectrum disorders (ASDs) in children 2–18 years old. A double-blind, randomized trial performed in several French centers using bumetanide treatment. Note that the CARS scale was reduced primarily and almost exclusively in children treated with bumetanide (blue columns) but not placebo with one exception (orange columns). Reducing CARS with 4 points or more is considered significant by EMA or FDA. From [41].

**Figure 4 cells-11-00396-f004:**
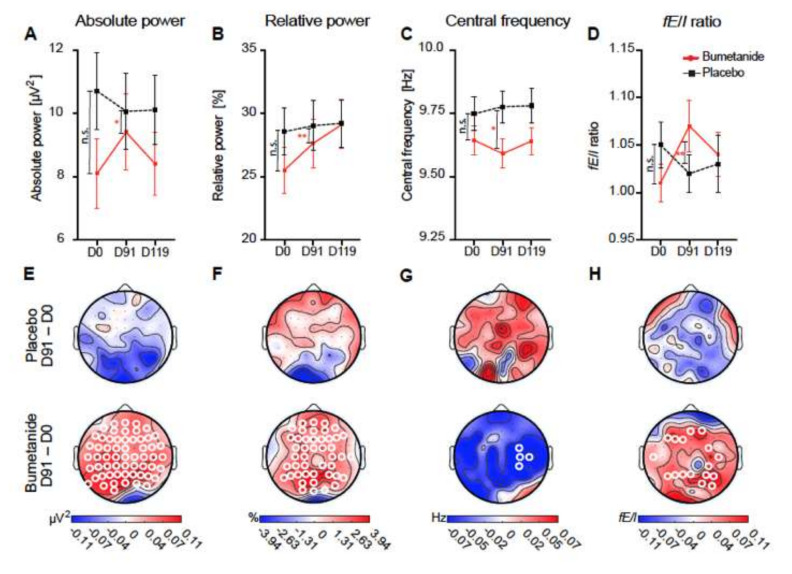
Bumetanide, but not placebo, affects alpha oscillations in children with ASDs: (**A**–**D**) whole-brain average EEG measures at treatment time points. Grand average topographies of the treatment on EEG measures. Significant changes are shown with white circles in (**E**–**H**); these are exclusively present in treated children, not placebo. This is observed in absolute EEG power, relative power, and fE/I ratio. Adapted with permission from [49].

**Table 1 cells-11-00396-t001:** Summary of the clinical trials using bumetanide to treat ASDs. Positive effects were observed in pilot or double-blind, randomized trials performed in France, China, Sweden, Great Britain, The Netherlands, and Tunisia. Two open trials evaluated eye-tracking and fMRI changes with bumetanide.

Country	N.	Age (y)	Rating Scale	Dose	Duration	End Points	Side Effects	References
China	119	3–6	CARS, ADOS, CGI, SRS	0.5 mg twice/day	3 months	Improvement in CARS score	mild (polyuria, hypokalemia)	[40]
Sweden	6	3–14	CARS	0.5 mg twice/day	4-12 weeks	Improvement in CARS score	mild (polyuria)	[42]
Netherland	92	7–15	SRS2	0.5 mg twice/day	3 months	Improvement in repetitive behavioral scale But not SRS2	mild (hypokalemia)	[45]
China	83	3–6	CARS, ADOS, CGI	0.5 mg twice/day	3 months	Reduction in CARS score, CGI-I	mild (polyuria)	[44]
Netherland	15	8–21	ABC-I (TSC)	0.5 mg twice/day	3 months	Improvement in ABC-I score EEG	mild (hypokalemia)	[50]
Tunisia	29	Average 7.9	ADI-R, CARS, CGI	0.1 mg/day	12 months	Improvement in CARS score	mild (hypokalemia)	[46]
France	9	Average 21.4	Eye tracking	1 mg/day	10 months	fMRI	None	[51]
France	88	2–18	CARS, SRS, CGI	0.5–2 mg twice/day	3 months	Improvement in CARS, CGI, SRS score	mild (hypokalemia)	[41]
China	60	Average 4.5	ABC, CARS, CGI	0.5 mg twice/day	3 months	Improvement in ABC, CARS, CGI score	None	[43]
France	7	Average 19.3	ADOS, fMRI emotion recognition	1 mg/day	10 months	Improvement performance for emotion recognition	mild (polyuria)	[51]
France	60	3–11	CARS, SRS, ADOS	1 mg/day	3 months	Improvement in CARS, ADOS score	mild (hypokalemia)	[38]
France	5	3–11	CARS, ABC, CGI, RDEG, RRB	1mg/day	3months	Improvement in CARS, CGI,	None	[39]

## Data Availability

Not applicable.

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
