# Peer review of "The GABA Polarity Shift and Bumetanide Treatment: Making Sense Requires Unbiased and Undogmatic Analysis"

_cells, 2022, doi:10.3390/cells11030396_

Round 1

Reviewer 1 Report

A summary of this review article: Development of cell chloride ion (Cl) homeostasis depends on developmental changes in NKCC1 and KCC2 expression. The type A GABA receptors (GABAARs) are GABA-gated channels permeable to Cl. The developmental switch of GABAAR action from excitation to inhibition in most central neurons is caused by changes in transmembrane Cl gradients, which are generated by cation-Cl co-transporters. The activity of Na+-K+-2Cl co-transporter 1 (NKCC1) accumulates the intracellular Cl concentration ([Cl]i) thus activating GABAARs leads to Cl efflux hence membrane depolarization in immature neurons. The subsequent phenotypic switch of GABAAR action from excitation to inhibition in mature neurons is due to upregulation of the Cl-extruder KCC2, which is a neuron-specific K+-Cl co-transporter, with or without downregulation of NKCC1 (depending on neurons in different brain regions), ultimately resulting in low [Cl]i levels and the hyperpolarizing action of GABAAR stimulation. Notably, the authors of this review article were the first to report the phenomenon that GABAergic inputs depolarize immature rat CA3 hippocampal neurones (Ben-ari Y, Cherubini E, Coraddetti , Gaiarsa, Jean-luc. 1989. “Journal of Physiology.” Journal of Physiology 5: 303–25). Later, Kaila, Rivera and colleagues showed that the developmental switch of GABAAR action results from the differential temporal expression of the cation-Cl- cotransporters NKCC1 and KCC2, involved in chloride uptake and extrusion, respectively (Rivera et al. 1999).

In this review article, the authors reply to some issues raised in the recent reviews by Kaila and colleagues (Virtanen et al. 2020; Löscher and Kaila 2021; Virtanen et al. 2021), debating that the downregulation of NKCC1 critically contributes to the developmental shift of GABAAR polarity in cortical neurons, and that the NKCC1 inhibitor Bumetanide has shown promising effects in clinical trials for treatment of some neurological diseases such as autism spectrum disorder (ASD). Such a review article may clarify some debate issues thus deepening the current understanding of development of Cl homeostasis in neuronal and non neuronal cells and broadening the view of potential use of Bumetanide for treating neurological illnesses.  

General comment: This article focuses specifically on the initial reports of the phenomenon of GABA depolarizing action in immature neurons and the molecular mechanisms of developmental shifts in [Cl]i, as well as the effects of bumetanide on ASD. Generally, the referenced papers are appropriate.

Specific comments:  

  1. Perhaps the sentence “Scientific debates are always welcome provided that they rely on facts not misinterpretations, misquotations and biasess (Reviewer -biases, or biasness, in lines 20-21)” can be omitted from the abstract.
  2. In lines 105-107, the authors state: “At birth, many rodent CA1 pyramidal neurons have little or no dendrites and therefore applying caged GABA on dendritic regions is difficult to interpret”. As shown by Tyzio et al. in 1999, most of the neurons (80%) do have an anlage of apical dendrite, although they do not show spontaneous or evoked PSCs. However, these “silent neurons” do express GABAA receptors. Does applying caged GABA on dendritic regionsit make sense?
  3. Lines-111-112: the authors state: “Single channel recordings remain the only reliable technique in particular in immature neurons” Did the authors mean cell-attached single channel recordings?
  4. Lines 226-228: the authors state: “… But even in the very unlikely case that the actions of Bumetanide were mediated exclusively by peripheral sites of action, this dogmatic view is ethically unacceptable, as ameliorating the life of infants with seizures or ASD remains an important aim”. Previous RT-PCR analysis indicated high expression of NKCC1, but not KCC2 cation/chloride transporter mRNAs in pituitary cells that express adrenocorticotropic hormone (ACTH), or growth hormone (GH). In voltage-clamped gramicidin-perforated cells, GABA induced dose-dependent increases in current amplitude that were inhibited by bicuculline and picrotoxin. J Physiol. 2008 Jul 1;586(13):3097-111. Regarding this notion, a previous study showed that serum levels of ACTH, GH and cortisol were significantly higher in subjects with autism than in controls. In addition, there was a significantly positive correlation between cortisol and ACTH levels in autism (Mol Autism. 2011 Oct 19; 2:16. doi: 10.1186/2040-2392-2-16). Based on these previous findings, the peripheral effect of Bumetanide should be considered.
  5. It is recently reported that in humans, bumetanide exposure was associated with a significantly lower AD prevalence in individuals over the age of 65 years in two electronic health record databases, suggesting the effectiveness of bumetanide in preventing AD (Nat Aging 1, 932–947 (2021). https://doi.org/10.1038/s43587-021-00122-7). In addition, a lately published meta review titled “Treatment Effect of Bumetanide in Children with Autism Spectrum Disorder: A Systematic Review and Meta-Analysis” (Front Psychiatry. 2021; 12: 751575) might be worthy of being cited in this review.

Author Response

Reply to the referees

We first would like to thank the referee for his (her) kind and detailed corrections and suggestions that we have incorporated in the revised Ms. Enclosed please find a detailed reply to the comments

Referee 1

In this review article, the authors reply to some issues raised in the recent reviews by Kaila and colleagues (Virtanen et al. 2020; Löscher and Kaila 2021; Virtanen et al. 2021), debating that the downregulation of NKCC1 critically contributes to the developmental shift of GABAAR polarity in cortical neurons, and that the NKCC1 inhibitor Bumetanide has shown promising effects in clinical trials for treatment of some neurological diseases such as autism spectrum disorder (ASD). Such a review article may clarify some debate issues thus deepening the current understanding of development of Cl homeostasis in neuronal and non neuronal cells and broadening the view of potential use of Bumetanide for treating neurological illnesses.  

General comment: This article focuses specifically on the initial reports of the phenomenon of GABA depolarizing action in immature neurons and the molecular mechanisms of developmental shifts in [Cl]i, as well as the effects of bumetanide on ASD. Generally, the referenced papers are appropriate.

Specific comments:  

  1. Perhaps the sentence “Scientific debates are always welcome provided that they rely on facts not misinterpretations, misquotations and biasess (Reviewer -biases, or biasness, in lines 20-21)” can be omitted from the abstract.

Thanks removed

  1. In lines 105-107, the authors state: “At birth, many rodent CA1 pyramidal neurons have little or no dendrites and therefore applying caged GABA on dendritic regions is difficult to interpret”. As shown by Tyzio et al. in 1999, most of the neurons (80%) do have an anlage of apical dendrite, although they do not show spontaneous or evoked PSCs. However, these “silent neurons” do express GABAA receptors. Does applying caged GABA on dendritic regions it make sense?

The problem here is that Kaila and co-workers claim that they are measuring DF GABA by measuring the flow of chloride between dendrites and cell body. If there are no dendrite -but at best an anlage- they are applying GABA in essence on the cell body that at this stage loses over 15mV when recorded by whole cell recordings. Therefore the interpretation of these observations remains doubtful.

  1. Lines-111-112: the authors state: “Single channel recordings remain the only reliable technique in particular in immature neurons” Did the authors mean cell-attached single channel recordings?

Yes of course thanks now clarified

  1. Lines 226-228: the authors state: “… But even in the very unlikely case that the actions of Bumetanide were mediated exclusively by peripheral sites of action, this dogmatic view is ethically unacceptable, as ameliorating the life of infants with seizures or ASD remains an important aim”. Previous RT-PCR analysis indicated high expression of NKCC1, but not KCC2 cation/chloride transporter mRNAs in pituitary cells that express adrenocorticotropic hormone (ACTH), or growth hormone (GH). In voltage-clamped gramicidin-perforated cells, GABA induced dose-dependent increases in current amplitude that were inhibited by bicuculline and picrotoxin. J Physiol. 2008 Jul 1;586(13):3097-111. Regarding this notion, a previous study showed that serum levels of ACTH, GH and cortisol were significantly higher in subjects with autism than in controls. In addition, there was a significantly positive correlation between cortisol and ACTH levels in autism (Mol Autism. 2011 Oct 19; 2:16. doi: 10.1186/2040-2392-2-16). Based on these previous findings, the peripheral effect of Bumetanide should be considered.

Thanks, we of course are not at all disregarding a peripheral action of bumetanide! the question is whether these are the only sites of actions -as Kaila and colleagues claim-we think that dissociating peripheral from central actions is illusory, the brain and the periphery are “ill” and impacted by the disease and the actions of Bumetanide are most likely peripheral and to some extent at least central. 

  1. It is recently reported that in humans, bumetanide exposure was associated with a significantly lower AD prevalence in individuals over the age of 65 years in two electronic health record databases, suggesting the effectiveness of bumetanide in preventing AD (Nat Aging 1, 932–947 (2021). https://doi.org/10.1038/s43587-021-00122-7). In addition, a lately published meta review titled “Treatment Effect of Bumetanide in Children with Autism Spectrum Disorder: A Systematic Review and Meta-Analysis” (Front Psychiatry. 2021; 12: 751575) might be worthy of being cited in this review.

Thanks, these papers were published after our submission, they are added, incidentally this paper on Alzheimer is simply fantastic in the way it was conducted and clearly shows the importance and relevance of bumetanide.

Reviewer 2 Report

The review article entitled “The GABA Polarity Shift and Bumetanide Treatment: Making Sense Requires Unbiased and Undogmatic Analysis” by Ben-Ari and Cherubini describes the issue of chloride equilibrium in neurons, in particular its developmental changes and correlations with neuropathologies. The main goal of the Authors is a comprehensive polemic with recent reviews that question the role of changes in the chloride balance in normal and pathological brain development. It is worth noting that both Authors are in fact the discoverers of the developmental changes in the concentration of chloride ions in neurons and the depolarizing properties of GABAergic transmission in the early stages of development (e.g. in neonatal mice).

In the opinion of the reviewer, the voice of professors Y. Ben-Ari and E. Cherubini is extremely important and necessary in the ongoing debate. In particular, abnormal changes in chloride balance may underlie many neurodevelopmental disorders such as Autism Spectrum Disorder and Rett Syndrome. Thus, a pharmacological intervention aimed at normalizing the chloride balance in neurons presents an interesting and promising option in the development of new therapies for the above-mentioned disorders. Importantly, preliminary clinical data support this line of research.

The review is, especially with provided figures, very informative and helps its readers to understand a thorough picture of how correct or abnormal developmental shift in neuronal chloride homeostasis affects the functioning of neuronal networks. This review/perspective would be suitable for researchers of inhibitory transmission in the brain but also in the clinical fields.

I have only couple minor comments as listed below:

  1. Line 47-48, ‘Earlier studies had observed a late maturation of inhibition but this study was the first to put together a global picture… ‘- please provide necessary citation of early paper/papers referred to in the first part of this sentence.
  2. Figures 1D and 1E are, in my opinion, hardly comprehensible to non-electrophysiologist and require more detailed description in the caption.
  3. Line 82-84, ‘In fact, the most reliable and direct evidence of the GABA shift stems from non-invasive single GABA channel recordings to determine EGABA and NMDA receptor mediated currents to determine Vrest.’ - please consider add appropriate citation.
  4. Line 150-151, ‘Yet, several hundreds of children with ASD have been and are treated successfully with Bumetanide’- please consider add citation to make this sentence more founded to the readers.
  5. Referring to Table 1. Does the Table 1 contain all clinical trials which used Bumetanide to assess its usefulness in ASD treatment or only the successful ones? If Table 1 contains only ‘positive clinical results’ please consider supplement the table with other clinical tests that did not show a statistically significant improvement.
  6. Line 174-176, ‘Also, atypical Event Related Potentials (ERP) were attenuated by Bumetanide providing an electrical partial signature of the amelioration’- please fill in the missing citation
  7. Line 211-212, ‘Incidentally, ototoxicity has never been observed in hundreds of children treated orally with Bumetanide, even during several years’ - please consider adding citations to strengthen the inference.
  8. Please consider citing prepared figures in the main text. In current version of the manuscript there is a detachment between figures and the text, and it will be harder to some readers to fully understand the importance of the details you intend to convey and sheer in the figures.
  9. Sentence in lines 241-243 ‘The actual trend os big pharma is to develop novel Bumetanide analogs and this attests to the importance of the pioneer works and trials we and others have made’ contains a typo and the term “big pharma” is rather colloquial.
  10. Line 257-258 ‘The next step would be to identify the sequels of the inaugurating insult and the deviatios it produces in brain construction’ should be redrafted due to the typo and not prime wording.

Author Response

Reviewer 2

We thank the referee for his (her) appreciation of this paper and his (her) comments that are fully integrated in the revised MS. The review article entitled “The GABA Polarity Shift and Bumetanide Treatment: Making Sense Requires Unbiased and Undogmatic Analysis” by Ben-Ari and Cherubini describes the issue of chloride equilibrium in neurons, in particular its developmental changes and correlations with neuropathologies. The main goal of the Authors is a comprehensive polemic with recent reviews that question the role of changes in the chloride balance in normal and pathological brain development. It is worth noting that both Authors are in fact the discoverers of the developmental changes in the concentration of chloride ions in neurons and the depolarizing properties of GABAergic transmission in the early stages of development (e.g. in neonatal mice).

In the opinion of the reviewer, the voice of professors Y. Ben-Ari and E. Cherubini is extremely important and necessary in the ongoing debate. In particular, abnormal changes in chloride balance may underlie many neurodevelopmental disorders such as Autism Spectrum Disorder and Rett Syndrome. Thus, a pharmacological intervention aimed at normalizing the chloride balance in neurons presents an interesting and promising option in the development of new therapies for the above-mentioned disorders. Importantly, preliminary clinical data support this line of research.

The review is, especially with provided figures, very informative and helps its readers to understand a thorough picture of how correct or abnormal developmental shift in neuronal chloride homeostasis affects the functioning of neuronal networks. This review/perspective would be suitable for researchers of inhibitory transmission in the brain but also in the clinical fields.

I have only couple minor comments as listed below:

  1. Line 47-48, ‘Earlier studies had observed a late maturation of inhibition but this study was the first to put together a global picture… ‘- please provide necessary citation of early paper/papers referred to in the first part of this sentence.

Thanks done

  1. Figures 1D and 1E are, in my opinion, hardly comprehensible to non-electrophysiologist and require more detailed description in the caption.

Thanks, caption rewritten

  1. Line 82-84, ‘In fact, the most reliable and direct evidence of the GABA shift stems from non-invasive single GABA channel recordings to determine EGABA and NMDA receptor mediated currents to determine Vrest.’ - please consider add appropriate citation.

Thanks done

Line 150-151, ‘Yet, several hundreds of children with ASD have been and are treated successfully with Bumetanide’- please consider add citation to make this sentence more founded to the readers.

Thanks added

  1. Referring to Table 1. Does the Table 1 contain all clinical trials which used Bumetanide to assess its usefulness in ASD treatment or only the successful ones? If Table 1 contains only ‘positive clinical results’ please consider supplement the table with other clinical tests that did not show a statistically significant improvement.

Thanks to the best of our knowledge the only study that did not find positive actions of Bumetanide in ASD is that of Bruining and colleagues. Yet, the authors insist even in that study that bumetanide does attenuate some syndrome of ASD and insist in the promising actions of bumetanide. Since then they have published another trial showin that Bumetanide does attenuate ASD in children/adolescents having special EEG features, in fact, the actions of bumetanide might be predicted by the EEG properties. These are now discussed

  1. Line 174-176, ‘Also, atypical Event Related Potentials (ERP) were attenuated by Bumetanide providing an electrical partial signature of the amelioration’- please fill in the missing citation

Thanks added

  1. Line 211-212, ‘Incidentally, ototoxicity has never been observed in hundreds of children treated orally with Bumetanide, even during several years’ - please consider adding citations to strengthen the inference.

Thanks it is difficult to provide a single reference to that effect We have corrected the statement to stress that “to the best of our knowledge there is not a single reference showing ototoxic actions of bumetanide following oral treatment with bumetanide”.

  1. Please consider citing prepared figures in the main text. In current version of the manuscript there is a detachment between figures and the text, and it will be harder to some readers to fully understand the importance of the details you intend to convey and sheer in the figures.

Thanks inserted and corrected.

  1. Sentence in lines 241-243 ‘The actual trend os big pharma is to develop novel Bumetanide analogs and this attests to the importance of the pioneer works and trials we and others have made’ contains a typo and the term “big pharma” is rather colloquial.

          Thanks corrected             

  1. Line 257-258 ‘The next step would be to identify the sequels of the inaugurating insult and the deviatios it produces in brain construction’ should be redrafted due to the typo and not prime wording.
  2. Thanks corrected
